# Fast and Accurate Stochastic Gradient Estimation

**Beidi Chen**
Rice University
Houston, Texas
beidi.chen@rice.edu

**Yingchen Xu**
Rice University
Houston, Texas
yx26@rice.edu

**Anshumali Shrivastava**
Rice University
Houston, Texas
anshumali@rice.edu

## Abstract

Stochastic Gradient Descent or SGD is the most popular optimization algorithm for large-scale problems. SGD estimates the gradient by uniform sampling with sample size one. There have been several other works that suggest faster epoch-wise convergence by using weighted non-uniform sampling for better gradient estimates. Unfortunately, the per-iteration cost of maintaining this adaptive distribution for gradient estimation is more than calculating the full gradient itself, which we call the chicken-and-the-egg loop. As a result, the false impression of faster convergence in iterations, in reality, leads to slower convergence in time. In this paper, we break this barrier by providing the first demonstration of a scheme, **L**ocality sensitive hashing (LSH) sampled Stochastic **G**radient **D**escent (LGD), which leads to superior gradient estimation while keeping the sampling cost per iteration similar to that of the uniform sampling. Such an algorithm is possible due to the sampling view of LSH, which came to light recently. As a consequence of superior and fast estimation, we reduce the running time of all existing gradient descent algorithms, that relies on gradient estimates including Adam, Ada-grad, etc. We demonstrate the effectiveness of our proposal with experiments on linear models as well as the non-linear BERT, which is a recent popular deep learning based language representation model.

## 1 Motivation

Stochastic gradient descent or commonly known as SGD is the most popular choice of optimization algorithm in large-scale setting for its computational efficiency. A typical interest in Machine Learning is to minimize the average loss function $f$ over the training data, with respect to the parameters $\theta$, i.e., the objective function of interest is

$$\theta^* = \arg\min_\theta F(\theta) = \arg\min_\theta \frac{1}{N} \sum_{i=1}^{N} f(x_i, \theta). \qquad (1)$$

Throughout the paper, our training data $D = \{x_i, \ y_i\}_{i=1}^{N}$ will have $N$ instances with $d$ dimensional features $x_i \in \mathbb{R}^d$ and labels $y_i$. The labels can be continuous real valued for regression problems. For classification problem, they will take value in a discrete set, i.e., $y_i \in \{1, \ 2, \cdots, \ K\}$. Typically, the function $f$ is convex, thus a Gradient Descent (GD) algorithm can achieve the global optimum. The objective function for least squares, $f(x_i, \theta) = (\theta \cdot x_i - y_i)^2$, used in regression setting is a classical example of $f$.

SGD [4] samples an instance $x_j$ uniformly from $N$ instances, and performs the gradient descent update:

$$\theta_t = \theta_{t-1} - \eta^t \nabla f(x_j, \theta_{t-1}), \qquad (2)$$

where $\eta^t$ is the step size at the $t^{th}$ iteration. The gradient $\nabla f(x_j, \theta_{t-1})$ is only evaluated on $x_j$, using the current $\theta_{t-1}$. It should be noted that a full gradient of the objective is given by the average

$\frac{1}{N} \sum_{i=1}^{N} \nabla f(x_i, \theta_{t-1})$. Thus, a uniformly sampled gradient $\nabla f(x_j, \theta_{t-1})$ is an unbiased estimator of the full gradient, i.e.,

$$\mathbb{E}(\nabla f(x_j, \theta_{t-1})) = \frac{1}{N} \sum_{i=1}^{N} \nabla f(x_i, \theta_{t-1}). \tag{3}$$

This is the key reason why, despite only using one sample, SGD still converges to the local minima, analogously to full gradient descent, provided $\eta^t$ is chosen properly [23, 4].

It is known that the convergence rate of SGD is slower than that of the full gradient descent [24]. Nevertheless, the cost of computing the full gradient requires $O(N)$ evaluations of $\nabla f$ compared to just $O(1)$ evaluation in SGD. Thus, with the cost of one epoch of full gradient descent, SGD can perform $O(N)$ epochs, which overcompensates the slow convergence (One epoch is one pass of the training data). Therefore, despite slow convergence rates, SGD is almost always the chosen algorithm in large-scale settings as the calculation of the full gradient in every epoch is prohibitively slow. Further improving SGD is still an active area of research. Any such improvement will directly speed up most of the state-of-the-art algorithms in machine learning.

The slower convergence of SGD in iterations is expected due to the poor estimation of the gradient (the average) by only sampling a single instance uniformly. Clearly, the variance of the one-sample estimator is high. As a consequence, there have been several efforts in finding better sampling strategies for estimating the gradients [31, 20, 2]. The key idea behind these methods is to replace sampling from a uniform distribution with sampling from a weighted distribution which leads towards a lower and even optimal variance.

However, obtaining the optimal weighted distribution is not a straightforward task, due to its correlation with the $L_2$ norm of the gradients. Therefore, whenever the parameters and the gradients change, the weighted distribution has to change. Unfortunately, as argued in [16, 22], all of these adaptive sampling methods for SGD, suffer from what we call the chicken-and-egg loop – adaptive sampling improves stochastic estimation but maintaining the required adaptive distribution will cost up to $O(N)$ per iteration, which is also the cost of computing the full gradient exactly (or at least not $O(1)$). Not surprisingly [19] showed another $O(N)$ scheme that improves the running time compared with SGD using $O(N)$ leverage scores [30] sampling. However, as noted $O(N)$ per iteration is prohibitive.

To the best of our knowledge, there does not exist any generic sampling scheme for adaptive gradient estimation, where the cost of maintaining and updating the distribution, per iteration, is $O(1)$ which is comparable to SGD. Our work provides first such sampling scheme utilizing the recent advances in sampling and unbiased estimation using Locality Sensitive Hashing [9, 10, 25, 26].

## 1.1 Related Work: Adaptive Sampling for SGD

For non-uniform sampling, we can sample each $x_i$ with an associated weight $w_i$. These $w_i$'s can be tuned to minimize the variance. It was first shown in [2], that sampling $x_i$ with probability in proportion to the $L_2$ norm (euclidean norm) of the gradient, i.e. $||\nabla f(x_i, \theta_{t-1})||_2$, leads to the optimal distribution that minimizes the variance. However, sampling $x_i$ with probability in proportion to $w_i = ||\nabla f(x_i, \theta_{t-1})||_2$, requires first computing all the $w_i$'s, which change in every iteration because $\theta_{t-1}$ gets updated. Therefore, maintaining the values of $w_i$'s is even costlier than computing the full gradient. [16] proposed to mitigate this overhead partially by exploiting additional side information such as the cluster structure of the data. Prior to the realization of optimal variance distribution, [31] and [20] proposed to sample a training instance with a probability proportional to the Lipschitz constant of the function $f(x_i, \theta_{t-1})$ or $\nabla f(x_i, \theta_{t-1})$ respectively. It is worth mentioning that before these works, a similar idea was used in designing importance sampling-based low-rank matrix approximation algorithms. The resulting sampling methods, known as leverage score sampling, are again proportional to the squared Euclidean norms of rows and columns of the underlying matrix [14]. Nevertheless, as argued, in [16], the cost of maintaining the distribution is prohibitive.

**The Chicken-and-Egg Loop:** In summary, to speed up the convergence of stochastic gradient descent, we need non-uniform sampling for better estimates (low variance) of the full gradient. Any interesting non-uniform sampling is dependent on the data and the parameter $\theta_t$ which changes in every iteration. Thus, maintaining the non-uniform distribution for estimation requires $O(N)$ computations to calculate the weight $w_i$, which is the same cost as computing it exactly. It is not even clear that there exists any sweet and adaptive distribution which breaks this computational

chicken-and-egg loop. We provide the first affirmative answer by giving an unusual distribution which is derived from probabilistic indexing based on locality sensitive hashing.

**Our Contributions:** In this work, we propose a novel LSH-based sampler, that breaks the aforementioned chicken-and-egg loop. Our algorithm, which we call **L**SH sampled Stochastic **G**radient **D**escent (LGD), are generated via hash lookups which have $O(1)$ cost. Moreover, the probability of selecting $x_i$ is provably adaptive. Therefore, the current gradient estimates is likely to have lower variance, compared to a single sample SGD, while the computational complexity of sampling is constant and of the order of SGD sampling cost. Furthermore, we demonstrate that LGD can be utilized to speed up any existing gradient-based optimization algorithm such as AdaGrad [15]. We also show the power of LGD with experiments on both linear and non-linear models.

As a direct consequence, we obtain a generic and efficient gradient descent algorithm which converges significantly faster than SGD, both in terms of iterations as well as running time. It should be noted that rapid iteration or epoch-wise convergence alone does not imply computational efficiency. For instance, Newtons method converges faster, epoch-wise, than any first-order gradient descent, but it is prohibitively slow in practice. The wall clock time or the amount of floating point operations performed to reach convergence should be the metric of consideration for useful conclusions.

**Accuracy Vs Running Time:** It is rare to see any fair (same computational setting) empirical comparisons of SGD with existing adaptive SGD schemes, which compare the improvement in accuracy with respect to running time on the same computational platform. Almost all methods compare accuracy with the number of epochs. It is unfair to SGD which can complete $O(N)$ updates at the computational cost (or running time) of one update for adaptive sampling schemes.

## 2 The LGD Algorithm

### 2.1 A Generic Framework for Efficient Gradient Estimation

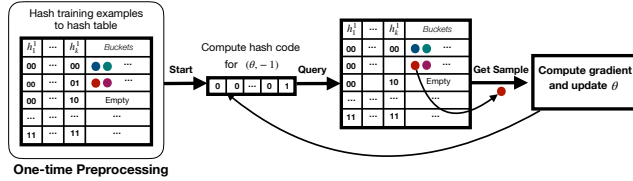

Figure 1: The work-flow of LGD Algorithm

Our algorithm leverages the efficient estimations using locality sensitive hashing, which usually beats random sampling estimators while keeping the sampling cost near-constant. We first provide the intuition of our proposal, and the analysis will follow. Figure 1 shows the complete work-flow of LGD algorithm. Consider least squares regression with loss function $\frac{1}{N}\sum_{i=1}^{N}(y_i - \theta_t \cdot x_i)^2$, where $\theta_t$ is the parameter in the $t^{th}$ iteration. The gradient is just like a partition function in classical discrete system. If we simply follow the procedures in [26], we can easily show a generic unbiased estimator via adaptive sampling. However, better sampling alternatives are possible.

Observing that the gradient, with respect to $\theta_t$ concerning $x_i$, is given by $2(y_i - \theta_t \cdot x_i)x_i$, the $L_2$ norm of the gradient can therefore be written as an absolute value of inner product.

$$\|\nabla f(x_i, \theta_t)\|_2 = \left|2(\theta_t \cdot x_i - y_i)\|x_i\|_2\right| = 2\left|[\theta_t, -1] \cdot [x_i\|x_i\|_2, y_i\|x_i\|_2]\right|, \quad (4)$$

where $[\theta_t, -1]$ is a vector concatenation of $\theta$ with $-1$. According to [16], $w_i^* = \frac{\|\nabla f(x_i, \theta_t)\|_2}{\sum_{j=1}^{N}\|\nabla f(x_j, \theta_t)\|_2}$ is also the optimal sampling weight for $x_i$. Therefore, if the data is normalized, we should sample $x_i$ in proportion to $w_{i*} = \left|[\theta_t, -1] \cdot [x_i, y_i]\right|$, i.e. large magnitude inner products should be sampled with higher probability.

As argued, such sampling process is expensive because $w_i^*$ changes with $\theta_t$. We address this issue by designing a sampling process that does not exactly sample with probability $w_i^*$ but instead samples from a different weighted distribution which is a monotonic function of $w_i^*$. Specifically, we sample from $w_i^{lsh} = f(w_i^*)$, where $f$ is some monotonic function. Before we describe the efficient sampling process, we first argue that a monotonic sampling is a good choice for gradient estimation. Figure 2 in the appendix helps visualize the relation among optimal weighted distribution (target), uniform sampling in SGD and adaptive sampling in LGD.

For any monotonic function $f$, the weighted distribution $w_i^{lsh} = f(w_i^*)$ is still adaptive and changes with $\theta_t$. Also, due to monotonicity, if the optimal sampling prefers $x_i$ over $x_j$ i.e. $w_i^* \geq w_j^*$, then

monotonic sampling will also have same preference, i.e., $w_i^{lsh} \geq w_j^{lsh}$. The key insight is that there

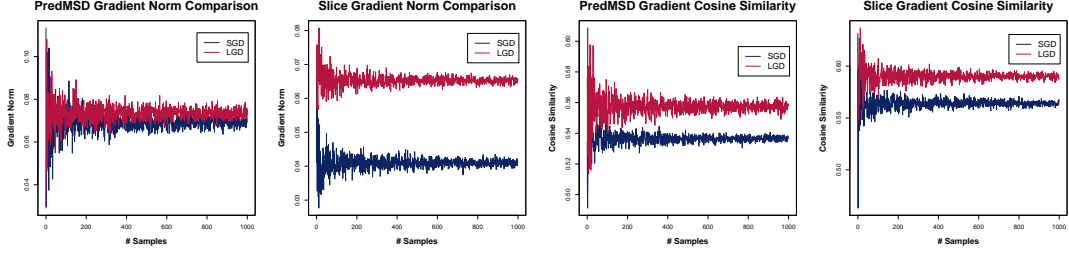

Figure 2: Subplots (a)(b) show the comparisons of the average (over number of samples) gradient $L_2$ norm of the points that LGD and SGD sampled. Subplots (d)(e) show the comparison of the cosine similarity between gradient estimated by LGD and the true gradient and the cosine similarity between gradient estimated by SGD and the true gradient.

are two quantities in the inner product (equation 4), $[\theta_t, -1]$ and $[x_i, y_i]$. With successive iteration, $[\theta_t, -1]$ changes while $[x_i, y_i]$ is fixed. Thus, it is possible to preprocess $[x_i, y_i]$ into hash tables (one time cost) and query with $[\theta_t, -1]$ for efficient and adaptive sampling. With every iteration, only the query changes to $[\theta_{t+1}, -1]$, but the hash tables remains the same. Few hash lookups are sufficient to sample $x_i$ for gradient estimation adaptively. Therefore, we only pay one-time preprocessing cost of building hash tables and few hash lookups, typically just one, in every iteration to get a sample for estimation.

---

**Algorithm 1:** assignment algorithm

**Input:** $H$ **(Hash functions),** $HT[][]$ **($L$ Hash Tables), K,** $Query$
$cp(x, Q)$ **is Pr(h(x)= h(Q)), under given LSH**
**Output: sampled data** $x$**, sampling probability** $p$
$l,\ S = 0$
**while** true **do**
    $ti = random(1, L)$
    $bucket = H(Query, ti)$ (table specific hash)
    **if** HT[ti][bucket] = empty **then**
        $l$++
    **end if**
    $S = |HT[ti][bucket]|$ (size of bucket)
    $x$ = randomly pick one element from $HT[ti][bucket]$
    break;
**end while**
$p = cp(x, Query)^K (1 - cp(x, Query)^K)^{l-1} \times \frac{1}{S}$
return $x,\ p$

---

There are a few more technical subtleties due to the absolute value of inner product $\big|[\theta_t, -1] \cdot [x_i, y_i]\big|$, rather than the inner product itself. However, the square of the absolute value of the inner product

$$\big|[\theta_t, -1] \cdot [x_i, y_i]\big|^2 = T([\theta_t, -1]) \cdot T([x_i, y_i]),$$

can also be written as an inner product as it is a quadratic kernel, and $T$ is the corresponding feature expansion transformation. Again square is monotonic function, and therefore, our sampling is still monotonic as composition of monotonic functions is monotonic. Thus, technically we hash $T([x_i, y_i])$ to create hash tables and the query at $t^{th}$ step is $T([\theta_t, -1])$. Once an $x_i$ is sampled via LSH sampling (Algorithm 1), we can precisely compute the probability of its sampling, i.e., $p_i$. It is not difficult to show that our estimation of full gradient is unbiased (Section 2.3).

## 2.2 Algorithm and Implementation Details

We first describe the detailed step of our gradient estimator in Algorithm 2. We also provide the sampling algorithm 1 with detail. Assume that we have access to the right LSH function $h$, and its collision probability expression $cp(x, y) = Pr(h(x) = h(y))$. For linear regression, we can use signed random projections, simhash [8], or MIPS hashing. With normalized data, simhash collision probability is $cp(x, y) = 1 - \frac{cos^{-1}(\frac{x \cdot y}{\|x\|_2 \|y\|_2})}{\pi}$, which is monotonic in the inner product. Furthermore, we centered the data we need to store in the LSH hash table to make the simhash query more efficient.

**LGD with Adaptive Learning Rate** The learning rate or step size $\eta$ in SGD is a one parameter approximation to the inverse of the Hessian (second order derivative) [5]. Time based (or step based) decay and exponential decay [29] have been empirically found to work well. Furthermore, [15] proposed the popular AdaGrad which is dimension specific adaptive learning rate based on first order gradient information. Although the methods mentioned above also help improve the convergence of

SGD by tweaking the learning rate, LGD is not an alternative but a complement to them. In LGD implementation, AdaGrad as well as those learning rate decay methods are customized options that can be used in conjunction.

**Algorithm 2:** LSH-Sampled Stochastic gradient Descent (LGD) Algorithm

1: **Input:** $D = x_i, y_i, N, \theta_0, \eta$
2: **Input: LSH Family $H$, parameters $K$, $L$**
3: **Output:** $\theta^*$
4: $HT$ = Get preprocessed training data vectors $x_{lsh}, y_{lsh}$ and then put $[x^i_{lsh}, y^i_{lsh}]$ into LSH Data structure.
5: Get $x'_{train}, y'_{train}$ from preprocessed data
6: $t = 0$
7: **while** $NotConverged$ **do**
8:    $x^i_{lsh}, p = Sample(H, HT, K, [\theta_t, -1])$ (Algorithm 1)
9:    Get $x^{i'}_{train}, y^{i'}_{train}$ from preprocessed data
10:   $\theta_{t+1} := \theta_t - \eta_t(\frac{\nabla f(x^{i'}_{train}, \theta_t)}{p \times N})$
11: **end while**
12: return $\theta^*$

**Running Time of Sampling** The computational cost of SGD sampling is merely a single random number generator. The cost of gradient update (equation 2) is one inner product, which is $d$ multiplications. If we want to design an adaptive sampling procedure that beats SGD, the sampling cost cannot be significantly larger than $d$ multiplications.

The cost of LGD sampling (Algorithm 1) is $K \times l$ hash computations followed by $l + 1$ random number generator, (1 extra for sampling from the bucket). Since the scheme works for any $K$, we can always choose $K$ small enough so that empty buckets are rare (see [26]). In all of our experiments, $K = 5$ for which $l$ is almost always 1. Thus, we require $K$ hash computations and only two random number generations. If we use very sparse random projections, then $K$ hash computations only require a constant

$\ll d$ multiplications. For example, in all our experiments we only need $\frac{d}{30}$ multiplication, in expectation, to get all the hashes using sparse projections. Therefore, our sampling cost is significantly less than $d$ multiplication which is the cost of gradient update. Using fast hash computation is critical for our method to work in practice.

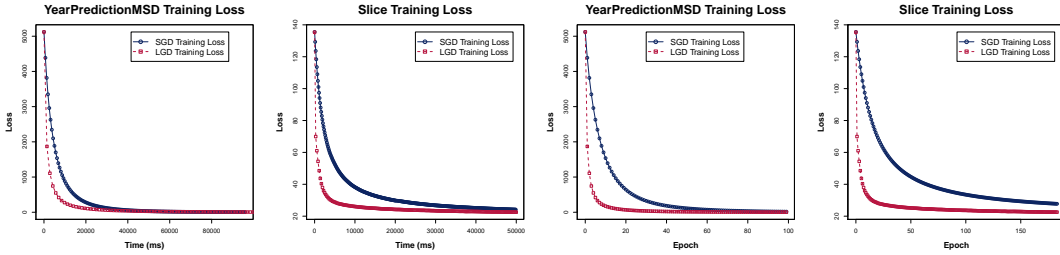

Figure 3: In subplots (a)(b), the comparisons of Wall clock training loss convergence are made between plain LGD (red lines) and plain SGD (blue lines). We can clearly see the big gap between them representing LGD converge faster than SGD even in time-wise. Subplots (d)(e) shows the results for same comparisons but in epoch-wise.

### 2.2.1   Near-Neighbor is Costlier than LSH-Sampling

It might be tempting to use approximate near-neighbor search with query $\theta_t$ to find $x_i$. Near-neighbor search has been used in past [12] to speed up coordinate descent. However, near-neighbor queries are expensive due to candidate generation and filtering. It is still sub-linear in $N$ (and not constant). Thus, even if we see epoch-wise faster convergence, iterations with a near-neighbor query would be orders of magnitude slower than a single SGD iteration. Moreover, the sampling probability of $x$ cannot be calculated for near-neighbor search which would cause bias in the gradient estimates.

It is important to note that although LSH is heavily used for near-neighbor search, in our case, we use it as a sampler. For efficient near neighbor search, $K$ and $L$ grow with $N$ [17]. In contrast, the sampling works for any $K$ and $l$ [1] as small as one leading to only approximately 1.5 times the cost of SGD iteration (see section 3). Efficient unbiased estimation is the key difference that makes sampling practical while near-neighbor query prohibitive. It is unlikely that a near-neighbor query would beat SGD in time, while sampling would.

## 2.3 Variance and Convergence Analysis

In section 1, we have discussed the convergence guarantees for SGD on convex functions under the assumptions of Lipschitz-continuous objective gradients with Lipschitz constant $L > 0$. Now we strengthen the importance of reducing the variance of for faster convergence rate. It is well-known that GD converges at a linear rate while SGD converges to a slower sublinear convergence rate. The key reason for the much slower rate is due to the relatively large variance of the SGD estimator. Specifically, assume that the variance of SGD is bounded in a relatively smaller manner, the expected decrease in the objective function yielded by the $t^{th}$ step is bounded by [6],

$$\mathbb{E}(f(\theta^{t+1})) - f(\theta^t) \leq -\eta_t \|\nabla f(x, \theta_t)\|^2 + \eta_t^2 \frac{L}{2}\mathbb{E}[\|\nabla f(x_i, \theta_t)\|^2] \tag{5}$$

If variance term $\mathbb{E}[\|\nabla f(x_i, \theta_t)\|^2] = 0$, then SGD would have had linear convergence rate with a constant step size similar to GD. However, due to the stochasticity introduced by the gradient estimation, the variance, smaller step size is chosen and thereby slowing down the convergence [5]. Clearly, lowering the variance of the estimator directly helps improve the convergence rate.

Therefore, in this section, we first prove that our estimator of the gradient is unbiased with bounded variance which is sufficient for convergence. We further argue about conditions under which LGD will have lower variance than SGD. Denote $S_b$ as the bucket that contains a set of samples which has the same hash value (same bucket) as the query and $x_m$ is the chosen sample in Algorithm 1. For simplicity we denote the query as $\theta_t$ and $p_i = cp(x_i, \theta_t)^K(1 - cp(x_i, \theta_t)^K)^{l-1}$ as the probability of $x_i$ belonging to that bucket.

**Theorem 1.** *The following expression is an unbiased estimator of the full gradient:*

$$Est = \frac{1}{N}\sum_{i=1}^{N} \mathbb{1}_{x_i \in S_b} \mathbb{1}_{(x_i = x_m | x_i \in S_b)} \frac{\nabla f(x_i, \theta_t) \cdot |S_b|}{p_i}, \qquad \mathbb{E}[Est] = \frac{1}{N}\sum_{i=1}^{N} \nabla f(x_i, \theta_t). \tag{6}$$

**Theorem 2.** *The Trace of the covariance of our estimator:*

$$Tr(\Sigma(Est)) = \frac{1}{N^2}\sum_{i=1}^{N} \frac{\|\nabla f(x_i, \theta_t)\|_2^2 \cdot \sum_{j=1}^{N} \frac{\mathbb{P}(x_i, x_j \in S_b)}{p_i}}{p_i} - \frac{1}{N^2}\|(\sum_{i=1}^{N} \nabla f(x_i, \theta_t))\|_2^2 \tag{7}$$

The trace of the covariance of LGD is the total variance of the descent direction. The variance can be minimized when the sampling probability of $x_i$ is proportional to the $L_2$-norm of the gradient we mentioned in Section 1.1. The intuition of the advantage of LGD estimator comes from sampling $x_i$ under a distribution monotonic to the optimal one. We first make a simple comparison of the variance of LGD with that of SGD theoretically and then in Section 3 and we would further empirically show the drastic superiority of LGD over SGD.

**Lemma 1.** *The Trace of the covariance of LGD's estimator is smaller than that of SGD's estimator if*

$$\frac{1}{N}\sum_{i=1}^{N} \frac{\|\nabla f(x_i, \theta_t)\|_2^2 \cdot \sum_{j=1}^{N} \frac{\mathbb{P}(x_i, x_j \in S_b)}{p_i}}{p_i} < \sum_{i=1}^{N} \|\nabla f(x_i, \theta_t)\|_2^2, \tag{8}$$

We analyze a simple case that if the data is uniformly distributed, such that every collision probability is the same. It is trivial to see that the trace of the covariance of LGD is exactly the same as SGD from equation 8. Intuitively, this happens when all the gradient norms are equal. Therefore, SGD would perform well if the data is uniform, but this is unlikely in practice.

Observe that when the gradients are large, $p_i$ is also large due to the monotonicity of $p_i$ with gradient norms. As a result, the term $\frac{\sum_{j=1}^{N} \frac{\mathbb{P}(x_i, x_j \in S_b)}{p_i}}{p_i}$ is likely to be much smaller than $N$ making the corresponding component of LHS (left hand side) smaller favoring LGD estimator. In a real scenario, with more of a power-law behavior, we have few large gradients, and most other gradients would be uniform in expectation. In such cases, We can expect LGD to have smaller variance. Rigorous characterization of distributions where LGD is better than SGD is hard due to correlations between gradients and collision probabilities $p$s as well as the size of buckets. [7] shows that such analysis is only possible under several query and data specific assumptions. A rigorous analysis in the gradient

descent settings where both query and data distribution changes in every iteration is left for the future work.

Here we provide the analysis based on assumptions on the data. We first upper bound the left side of equation 8 by,

$$\frac{1}{N} \sum_{i=1}^{N} \frac{\|\nabla f(x_i, \theta_t)\|_2^2 \cdot \sum_{j=1}^{N} \frac{\mathbb{P}(x_i, x_j \in S_b)}{p_i}}{p_i} \leq \sum_{i=1}^{N} \|\nabla f(x_i, \theta_t)\|_2^2 \cdot \frac{\sum_{j=1}^{N} p_j}{p_i^2 N} \qquad (9)$$

Assume the normalized collision probability follows Pareto distribution [3], which is a power-law probability distribution. If $X$ is a random variable with a Pareto (Type I) distribution, then the probability that $X$ is greater than some number $x$, is $Pr(X > x) = \begin{cases} (\frac{x_m}{x})^{\alpha}, & \text{if } x > x_m \\ 1, & \text{if } x \leq x_m \end{cases}$ where $x_m$ is the minimum possible value of X, and $\alpha$ is a positive parameter. Then $\frac{\sum_{j=1}^{N} p_j}{N}$ (mean) is $\mu_p = \frac{\alpha x_m}{\alpha - 1}$. Assume $p_i$ is sorted in descending order and let first separate the right side of equation 9 into two parts, $\sum_{i=1}^{k} \|\nabla f(x_i, \theta_t)\|_2^2 \cdot \frac{\mu_p}{p_i^2} + \sum_{i=k+1}^{N} \|\nabla f(x_i, \theta_t)\|_2^2 \cdot \frac{\mu_p}{p_i^2}$, where k is the index that separates the summation based on $\mu_p \leq p_i^2$ or $\mu_p > p_i^2$. Then we equation 8 becomes,

$$\sum_{i=1}^{k} \|\nabla f(x_i, \theta_t)\|_2^2 \cdot (1 - \frac{\mu_p}{p_i^2}) > \sum_{i=k+1}^{N} \|\nabla f(x_i, \theta_t)\|_2^2 \cdot (\frac{\mu_p}{p_i^2} - 1), \qquad (10)$$

making lemma 1 a reasonable condition if the distribution of the gradient norm also follows power-law. Because the large gradient norm terms will be on the LHS and the small gradient terms are on the RHS and under power-law assumption the small gradient norms terms drop off extremely fast.

In practice, we can tune parameter $K$ for our hashing scheme in LGD, which controls the values of $p_i$. With this tuning, we achieve better controls over relative decays of $p_i$ leading to more possibilities of better variance. Recall that the collision probability $p_i = cp(x_i, \theta_t)^K (1 - cp(x_i, \theta_t)^K)^{l-1}$. Note that $l$ here, according to Algorithm 1 is the number of tables that have been utilized by the sampling process. In most practical cases and also in our experiment, $K$ and $l$ are relatively small. $L$, which is the total number of hash tables, should be large to ensure enough independence across samples, but it does not contribute to the sampling time (See Alg. 1). Overall, our experiments show that LGD is efficient and generally achieves smaller variance than SGD by setting small enough values of $K$ and $l$ making the sampling process as efficient as SGD.

**LGD for Logistic Regression** We can derive a similar form of LGD for logistic regression. Noted that the label $y_i \in \{-1, +1\}$. The loss function of logistic regression can be written as, $L(\theta_t) = \frac{1}{N} \sum_{i=1}^{N} ln(1 + e^{-y_i \theta_t x_i})$, s where the $l_2$ norm of the gradient can be derived as,

$$\|\nabla L(\theta_t)_i\|_2 = \frac{\|x_i\|_2}{e^{y_i \theta_t x_i} + 1} = \frac{1}{e^{y_i \theta_t x_i} + 1}, \qquad (11)$$

when $x_i$ is normalized to have unit norm. Similar to linear regression, we get two quantities in the inner product, $y_i \cdot x_i$ and $-\theta_t$. The inner product is monotonic to $\frac{1}{e^{y_i \theta_t x_i} + 1}$, which is the $l_2$ norm of the gradient. To apply our LGD framework for estimating the gradient, we can preprocess $y_i \cdot x_i$ into hash tables and query with $-\theta_t$ for efficient and adaptive sampling.

# 3 Experiments

Linear regression is a basic and commonly used supervised machine learning algorithm for prediction. Deep learning models recently become popular for their state-of-the-art performance on Natural Language Processing (NLP) and also Computer Vision tasks. Therefore, we chose both linear regression and deep learning models as the target experiment tasks to examine the effectiveness of our algorithm. We follow the following four steps: (1) Compare the quality of samples retrieved by LGD and that of samples retrieved by SGD. According to Section 2.1, high quality samples have larger gradient $L_2$ norm. (2) Compare the convergence of linear regression task in time using SGD and LGD. (3) Compare the convergence of linear regression task in time using SGD with AdaGrad and LGD with AdaGrad. (4) Compare the epoch-wise convergence of NLP tasks between SGD and LGD in with BERT [11].

**Dataset:** We used three large regression, YearPredictionMSD [18],Slice [18], UJIIndoorLoc [27], and two NLP benchmarks, MRPC [13], RTE [28]. The details are shown in Table 4 and Appendix.

## 3.1 Linear Regression Tasks

[2] Three regression datasets were preprocessed as described in Section 2.2. Note that for all the experiments, the choice of the gradient decent algorithm was the same. For both SGD and LGD, the only difference in the gradient algorithm was the gradient estimator. For SGD, a random sampling estimator was used, while for LGD, the estimator used the adaptive estimator. We used fixed values $K = 5$ and $L = 100$ for all the datasets. $l$ is the number of hash tables that have been searched before landing in a non-empty bucket in a query. In our experiments $l$ is almost always as low as 1. $L$ only affects preprocessing but not sampling. Our hash function was simhash (or signed random projections) and we used sparse random projections with sparsity $\frac{1}{30}$ for speed. We know that epoch-wise convergence is not a true indicator of speed as it hides per epoch computation. Our main focus is convergence with running time, which is a better indicator of computational efficiency.

To the best of our knowledge, there is no other adaptive estimation baseline, where the cost of sampling per iteration is less than linear $O(N)$. Since our primary focus would be on wall clock speedup, no $O(N)$ estimation method would be able to outperform $O(1)$ SGD (and LGD) estimates on the same platform. From section 2.2.1, even methods

Figure 4: Statistics Information for Datasets

| DATA SET | TRAINING | TESTING | DIMENSION |
|---|---|---|---|
| YEARMSD | 463,715 | 51,630 | 90 |
| SLICE | 53,500 | 42,800 | 74 |
| UJIINDOORLOC | 10,534 | 10,534 | 529 |
| MRPC | 3669 | 409 | N/A |
| RTE | 2491 | 278 | N/A |

requiring a near-neighbor query would be too costly (orders of magnitude) to outperform SGD from computational perspective.

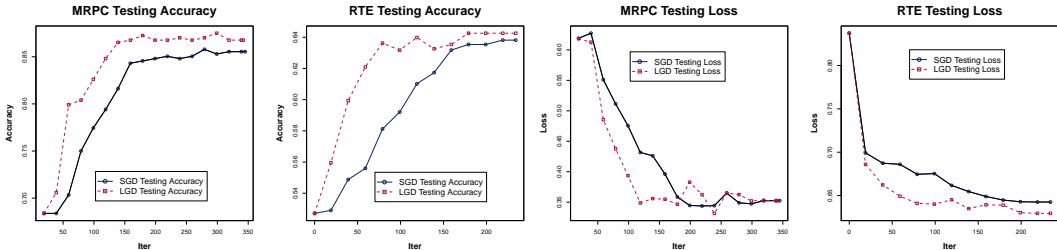

Figure 5: In subplots (a)(b), the comparisons of epoch-wise testing accuracy convergence are made between LGD (red lines) and SGD (blue lines) separately in two NLP benchmarks. We can see the big gap between them representing LGD converge faster than SGD. Subplots (c)(d) shows similar comparison over testing loss.

**LGD, SGD vs. True Gradient** In the first experiment, as a sanity check, we first verify weather LGD samples data point with probability monotonic to $L_2$ norm of the gradient mentioned in section 2.1. In order to do that, we freeze the optimization at an intermediate iteration and use the $\theta$ at that moment to sample data points with LGD as well as SGD to compute gradient $L_2$ norm separately. We observe that if freezing at the beginning iterations, the difference of average gradient norm between LGD and SGD samples is not obvious. This is not surprising because model $\theta$ is initialized randomly. To visualize the quality difference of SGD and LGD samples more clearly, we choose to freeze after $\frac{1}{4}$ epoch of cold start. The upper three plots in Figure 2 show the comparison of the sampled gradient norm of LGD and SGD. X-axis represents the number of samples that we averaged in the above process. It is obvious that LGD sampled points have larger gradient norm than SGD ones consistently across all three datasets.

In addition, we also do a sanity check that if empirically, the chosen sample from LGD get better estimation of the true gradient direction than that of SGD. Again, we freeze the program at an intermediate iteration like the experiments above. Then we compute the angular similarity of full gradient (average over the training data) direction with both LGD and SGD gradient direction, where, $Similarity = 1 - \frac{cos^{-1}\frac{x \cdot y}{\|x\|_2 \|y\|_2}}{\pi}$. From the right two plots in Figure 2, we can see that in average,

LGD estimated gradient has smaller angle (more aligned) to true gradient than SGD estimated gradient.The variance of both norm and cosine similarity reduce when averaging them over samples as shown in plots.

**LGD vs. SGD** In this section, we compare vanilla SGD with LGD, i.e., we use simple SGD with fixed learning rate. This basic experiment aims to demonstrate the performance of pure LGD and SGD without involving other factors like $L_1/L_2$ regularization on linear regression task. In such a way, we can quantify the superiority of LGD more easily. We tried a sweep of initial step size from $1e^{-5}$ to $1e^{-1}$ and choose the one that will lead to convergence with LGD and SGD. Figure 3 shows the decrease in the squared loss error with epochs. Blue lines represent SGD and red lines represent LGD. It is obvious that LGD converges much faster than SGD in both training and testing loss comparisons. This is not surprising with the claims in

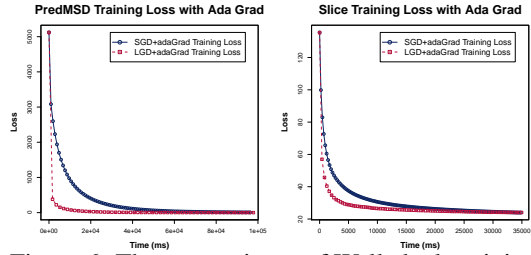

Figure 6: The comparisons of Wall clock training loss convergence are made between LGD+adaGrad and SGD+adaGrad separately in three datasets. We can again see the similar gap between them representing LGD converge faster than SGD in time-wise. Epoch-wise comparisons are in appendix.

Section 2.2 and theoretical proof in Section 2.3. Since LGD uses slightly more computations per epoch than SGD does, it is hard to defend if LGD gains enough benefits simply from the epoch-wise comparisons. We therefore also show the decrease in error with wall clock time also in figure 3. Wall clock time is the actual quantification of speedups. Again, on every single dataset, LGD shows faster time-wise convergence as well.

As argued in section 1.1, our LGD algorithm is complimentary to any gradient-based optimization algorithm. We repeated the first experiment but using AdaGrad [15] instead of plain SGD. Figure 6 shows running time comparisons on LGD and SGD training convergence. The trends as expected are similar to those of LGD vs. SGD. LGD with AdaGrad outperforms AdaGrad (SGD) estimates of gradients both epoch-wise and time-wise.

## 3.2 BERT Tasks

BERT [11], a recent popular language representation model, is designed to pre-train deep bidirectional representations that can be fine-tuned jointly with just one additional layer to create state-of-the-art models for various tasks. To strengthen the power of LGD, we adapted LGD in BERT for several natural language processing (NLP) tasks. The implementation details were included in the appendix. We used two popular benchmarks in NLP, MRPC and RTE, and replicated the same experiments setting in BERT paper. For the pre-trained model, we chose $BERT_{base}$ because it performs more stable for such smaller downstream tasks. For each task, we ran fine-tunings for 3 epochs with batch size 32 and used Adam optimizer with initial learning rates $2e$. As for LSH parameter, we chose $K = 7$, $L = 10$. Results are presented in Figure 5. We show that LGD outperformed SGD in epoch-wise convergence on both tasks with a substantial margin. It is encouraging because in the previous section, we have shown that even with the hashing overhead, LGD leads to faster time-wise convergence. We do not explore the time-wise convergence comparison between LGD and SGD in current tasks because BERT is implemented in Tensorflow [1] and Pytorch [21] on GPU. We currently only have the CPU implementation of LSH. Therefore running LGD algorithm on BERT creates an extra overhead of switching between GPUs and CPUs. An efficient GPU implementation of LGD can be an independent research interest for future work. This section is to demonstrate the power of LGD in non-linear models.

## 4    Conclusion

In this paper, we proposed a novel LSH-based sampler with a reduction to the gradient estimation variance. We achieved it by sampling with probability proportional to the $L_2$ norm of the instances gradients leading to an optimal distribution that minimizes the variance of estimation. More remarkably, LGD is as computationally efficient as SGD but achieves faster convergence not only epoch-wise but also time-wise.

## Acknowledgments

We thank the reviewers for their valuable comments. We also thank Ben Benjamin Coleman for the helpful discussions. The work was supported by NSF-1652131, NSF-BIGDATA 1838177, AFOSR-YIPFA9550-18-1-0152, Amazon Research Award, and ONR BRC grant for Randomized Numerical Linear Algebra.

## Footnotes

[1] L represents the number of hash tables but l represents the number of hash tables used in one query

[2]Note that in the experiments, we show the plots of two datasets and the third one is in the appendix.

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
