[Supplementary Material]

# Fast and Accurate Stochastic Gradient Estimation

**Beidi Chen**
Rice University
Houston, Texas
beidi.chen@rice.edu

**Yingchen Xu**
Rice University
Houston, Texas
yx26@rice.edu

**Anshumali Shrivastava**
Rice University
Houston, Texas
anshumali@rice.edu

## 1 More details for LSH

In this section, we first describe a recent advancement in the theory of sampling and estimation using locality sensitive hashing (LSH) (Indyk & Motwani, 1998b) which will be heavily used in our proposal. Before we get into the details of sampling, let us revise the two-decade-old theory of LSH.

### 1.1 Locality Sensitive Hashing (LSH)

This section briefly reviews LSH for large-scale nearest-neighbor search. Please refer to Indyk & Motwani (1998a); Indyk & Woodruff (2006) for more details.

A favorite sub-linear time algorithm for approximating the nearest-neighbor search uses the underlying theory of *Locality Sensitive Hashing* Indyk & Motwani (1998a). LSH is a family of functions, with the property that similar input objects in the domain of these functions have a higher probability of colliding in the range space than non-similar ones. In formal terms, consider $\mathcal{H}$ a family of hash functions mapping $\mathbb{R}$ to some set $\mathcal{S}$.

**Definition 1** (**LSH Family**). *A family $\mathcal{H}$ is called*
$(S_0, cS_0, p_1, p_2)$-*sensitive if for any two point $x, y \in \mathbb{R}$ and $h$ chosen uniformly from $\mathcal{H}$ satisfies the following:*

- *if $Sim(x, y) \geq S_0$ then $Pr(h(x) = h(y)) \geq p_1$*

- *if $Sim(x, y) \leq cS_0$ then $Pr(h(x) = h(y)) \leq p_2$*

For approximate nearest neighbor search typically, $p_1 > p_2$ and $c < 1$ is needed. LSH allows us to construct data structures that give provably efficient query time algorithms for approximate nearest-neighbor problem with the associated similarity measure.

Figure 1: Example of LSH hash tables

One sufficient condition for a hash family $\mathcal{H}$ to be a LSH family is that the **collision probability** $Pr_{\mathcal{H}}(h(x) = h(y))$ is a monotonically increasing function of the similarity $Sim$, i.e.

$$Pr_{\mathcal{H}}(h(x) = h(y)) = \mathcal{M}(Sim(x, y)), \tag{1}$$

where $\mathcal{M}$ is a monotonically increasing function. Essentially, similar items are more likely to collide with each other under the same hash fingerprint. In fact, most of the popular known LSH family, such

as SimHash (Section 1.2), actually satisfies this stronger property. It can be noted that Equation 1 automatically guarantees the two required conditions in Definition 1 for any $S_0$ and $c < 1$.

It was shown Indyk & Motwani (1998a) that having a LSH family for a given similarity measure is sufficient for efficiently solving a nearest-neighbor search problem in sub-linear time.

The algorithm uses two parameters, $(K, L)$. We construct $L$ independent hash tables from the collection $\mathcal{C}$. Each hash table has a meta-hash function $H$ that is formed by concatenating $K$ random independent hash functions from $\mathcal{F}$. Given a query, we collect one bucket from each hash table and return the union of $L$ buckets. Figure 1 shows the visualization of the hash tables. Intuitively, the meta-hash function makes the buckets sparse and reduces the number of false positives, because only valid nearest-neighbor items are likely to match all $K$ hash values for a given query. The union of the $L$ buckets decreases the number of false negatives by increasing the number of potential buckets that could hold valid nearest-neighbor items.

The candidate generation algorithm works in two phases [See (Spring & Shrivastava, 2017) for details]:

1. **Pre-processing Phase:** We construct $L$ hash tables from the data by storing all elements $x \in \mathcal{C}$. We only store pointers to the vector in the hash tables because storing whole data vectors is very memory inefficient.

2. **Query Phase:** Given a query $Q$; we will search for its nearest-neighbors. We report the union from all of the buckets collected from the $L$ hash tables. Note, we do not scan all of the elements in $\mathcal{C}$, we only probe $L$ different buckets, one bucket for each hash table.

After generating the set of potential candidates, the nearest-neighbor is computed by comparing the distance between each item in the candidate set and the query.

Figure 2: shows the optimal weighted distribution (target), uniform sampling in SGD and adaptive sampling in LGD.

## 1.2 Popular LSH: Signed Random Projections (SimHash) and Cosine Similarity

SimHash is a popular LSH, which originates from the concept of **Signed Random Projections (SRP)** Charikar (2002); Rajaraman & Ullman (2011); Henzinger (2006) for the cosine similarity measure. Given a vector $x$, SRP utilizes a random $w$ vector with each component generated from i.i.d. normal, i.e., $w_i \sim N(0, 1)$, and only stores the sign of the projection. Formally SimHash is given by

$$h_w^{sign}(x) = sign(w^T x). \tag{2}$$

It was shown in the seminal work Goemans & Williamson (1995) that collision probability under SRP satisfies the following equation:

$$Pr(h_w^{sign}(x) = h_w^{sign}(y)) = 1 - \frac{\theta}{\pi}, \tag{3}$$

where $\theta = cos^{-1}\left(\frac{x^T y}{||x||_2 \cdot ||y||_2}\right)$. The term $\frac{x^T y}{||x||_2 \cdot ||y||_2}$ is the cosine similarity. There is a variant of SimHash where, instead of $w_i \sim N(0, 1)$, we choose each $w_i$ independently as either +1 or -1 with probability $\frac{1}{2}$. It is known that this variant performs similarly to the one with $w \sim N(0, 1)$ Rajaraman & Ullman (2011). Since $1 - \frac{\theta}{\pi}$ is monotonic to cosine similarity $\mathcal{S}$, it is a valid LSH.

Figure 3: Norm and cosine similarity comparisons of LGD and SGD gradient estimation. Subplots (a)(b)(c) show the comparisons of the average (over number of samples) gradient $L_2$ norm of the points that LGD (red lines) and SGD sampled (blue lines). As argued before, LGD samples with probability monotonic to $L_2$ norm of the gradients while SGD samples uniformly. It matches with the results shown in the plots that LGD queries points with larger gradient than SGD does. Subplots (d)(e)(f) show the comparison of the cosine similarity between gradient estimated by LGD and the true gradient and the cosine similarity between gradient estimated by SGD and the true gradient. Note that the variance of both norm and cosine similarity reduce when we average over more samples.

## 2 More Algorithm Details

### 2.1 Sampling from Hash Tables

$L$ independent hash tables are constructed during the initialization phase, which is a one-time cost. During the sampling phase, LGD does not collect the union of samples from buckets in all L hash tables for efficiency. Because L is usually large to guarantee the randomness involved in the hash tables so that all the training samples can be covered. Instead, in each iteration, LGD randomly selects a hash table to retrieve a sample. If the bucket is empty, LGD will keep searching in L tables until it finds a non-empty bucket.

### 2.2 LGD for Mini-batch

There is a compromise between computing the batch gradient and the gradient at a single sample called mini-batch gradient descent Li et al. (2014). It computes the gradient against more than one training sample at each step, and the batch size is also a hyper-parameter that can be tuned for better performance. It results in smoother convergence, as the gradient computed at each step is averaged over more training examples. This trick can also be applied to LGD without affecting its efficiency. We can see that according to the hash codes of the query, a random sample from the matching bucket in a random hash table is chosen for plain LGD. Therefore, if a mini-batch of $m$ samples is needed for every iteration and the first matching bucket LGD find has $n$ points, in our implementation, LGD can sample $m$ examples from that bucket when $m < n$. Otherwise, LGD will continue sampling

Figure 4: In subplots (a)(b)(c), the comparisons of Wall clock training loss convergence are made between plain LGD (red lines) and plain SGD (blue lines) separately on three datasets. We can clearly see the big gap between them representing LGD converge faster than SGD even in time wise. Subplots (d)(e)(f) shows the results for same comparisons but in epoch wise. We can see that LGD converges even faster than SGD which is not surprising because LGD costs a bit more time than SGD does in every iteration.

from matching buckets in other hash tables until $n$ samples have been collected or all hash tables have been visited.

# 3 Variance Analysis Proofs

**Theorem 1.** *In this section, we first prove that our estimator of the gradient is unbiased with lower variance than SGD for most real datasets. Denote $S_b$ as the bucket that contains a set of samples which has the same hash value (same bucket) as the query and $x_m$ is the chosen sample in Algorithm 2. For simplicity we denote the query as $\theta_t$ and $p_i = cp(x_i, \theta_t)^K (1 - cp(x_i, \theta_t)^K)^{l-1}$ as the probability of $x_i$ belonging to that bucket.*

$$Est = \frac{1}{N} \sum_{i=1}^{N} \mathbb{1}_{x_i \in S_b} \mathbb{1}_{(x_i = x_m | x_i \in S_b)} \frac{\nabla f(x_i, \theta_t) \cdot |S_b|}{p_i}$$

$$\mathbb{E}[Est] = \frac{1}{N} \sum_{i=1}^{N} \nabla f(x_i, \theta_t)$$

*Proof.*

$$\mathbb{E}[\mathbb{1}_{x_i \in S_b}] = p_i, \quad and \quad \mathbb{E}[\mathbb{1}_{x_i = x_m | x_i \in S_b}] = \frac{1}{|S_b|}.$$

Also note that

$$\mathbb{E}[\mathbb{1}_{x_i \in S_b} \mathbb{1}_{x_i = x_m | x_i \in S_b}] = \mathbb{E}[\mathbb{1}_{x_i \in S_b}] \mathbb{E}[\mathbb{1}_{x_i = x_m | x_i \in S_b}].$$

Figure 5: In subplots (a)(b)(c), the comparisons of Wall clock testing loss convergence are made between plain LGD (red lines) and plain SGD (blue lines) on three datasets. We can see the gap between them representing LGD converge faster than SGD even in time wise. Subplots (d)(e)(f) shows the results for same comparisons but in epoch wise. We can see that LGD converges even faster than SGD.

Then,

$$
\begin{aligned}
\mathbb{E}[Est] &= \frac{1}{N}\mathbb{E}[\sum_{i=1}^{N}\mathbb{1}_{x_i \in S_b}\mathbb{1}_{x_i=x_m|x_i \in S_b}\frac{\nabla f(x_i,\theta_t)\cdot|S_b|}{p_i}]\\
&= \frac{1}{N}\sum_{i=1}^{N}\mathbb{E}[\mathbb{1}_{x_i \in S_b}\mathbb{1}_{x_i=x_m|x_i \in S_b}]\cdot\mathbb{E}[\frac{\nabla f(x_i,\theta_t)\cdot|S_b|}{p_i}]\\
&= \frac{1}{N}\sum_{i=1}^{N}p_i\cdot\frac{1}{|S_b|}\frac{\nabla f(x_i,\theta_t)\cdot|S_b|}{p_i}\\
&= \frac{1}{N}\sum_{i=1}^{N}\nabla f(x_i,\theta_t)
\end{aligned}
$$

$\square$

**Theorem 2.** *The Trace of the covariance of our estimator is:*

$$
Tr(\Sigma(Est)) = \frac{1}{N^2}\sum_{i=1}^{N}\frac{\|\nabla f(x_i,\theta_t)\|_2^2\cdot\mathbb{E}(|S_b|)}{p_i} - \frac{1}{N^2}(\sum_{i=1}^{N}\|\nabla f(x_i,\theta_t)\|_2)^2.
$$

*Proof.*

$$
Tr(\Sigma(Est) = \mathbb{E}[Est^T Est] - \mathbb{E}[Est]^T\mathbb{E}[Est]
$$

| ((a)) YearPredictionMSD | ((b)) Slice | ((c)) UJIIndoorLoc |
| ((d)) YearPredictionMSD | ((e)) Slice | ((f)) UJIIndoorLoc |

Figure 6: In subplots (a)(b)(c), the comparisons of Wall clock training loss convergence are made between LGD+adaGrad (red lines) and SGD+adaGrad (blue lines) separately in three datasets. We can again see the similar gap between them representing LGD converge faster than SGD in time wise. Subplots (d)(e)(f) show the results for same comparisons but in epoch wise. We can see that LGD converges even faster than SGD.

$$Est^T Est = \frac{1}{N^2} \sum_{i,j}^{N} \mathbb{1}_{x_i \in S_b} \mathbb{1}_{x_j \in S_b} \mathbb{1}_{x_i = x_m | x_i \in S_b} \mathbb{1}_{x_j = x_m | x_j \in S_b} \frac{\nabla f(x_i, \theta_t) \cdot \nabla f(x_j, \theta_t) \cdot |S_b|^2}{p_i \cdot p_j}$$

$$= \frac{1}{N^2} \sum_{i}^{N} \mathbb{1}_{x_i \in S_b} \cdot \mathbb{1}_{x_i = x_m | x_i \in S_b} \frac{(\nabla \|f(x_i, \theta_t)\|)\|_2^2 \cdot |S_b|^2}{p_i^2}$$

$$\mathbb{E}[Est^T Est] = \frac{1}{N^2} \sum_{i}^{N} \frac{(\|\nabla f(x_i, \theta_t)\|)\|_2^2 \cdot \mathbb{E}(|S_b| \,|\, x_i \in S_b))}{p_i}$$

$$Tr(\Sigma(Est)) = \frac{1}{N^2} \sum_{i=1}^{N} \frac{\|\nabla f(x_i, \theta_t)\|_2^2 \cdot \mathbb{E}(|S_b| \,|\, x_i \in S_b))}{p_i} - \frac{1}{N^2} (\sum_{i=1}^{N} \nabla \|f(x_i, \theta_t)\|_2)^2 \tag{4}$$

$$= \frac{1}{N^2} \sum_{i=1}^{N} \frac{\|\nabla f(x_i, \theta_t)\|_2^2 \cdot \sum_{j=1}^{N} \mathbb{P}(x_i, x_j \in S_b)}{p_i^2} - \frac{1}{N^2} \|(\sum_{i=1}^{N} \nabla f(x_i, \theta_t))\|_2^2 \tag{5}$$

$\square$

**Lemma 1.** *The Trace of the covariance of LSD's estimator is smaller than that of SGD's estimator if*

$$\frac{1}{N} \sum_{i=1}^{N} \frac{\|\nabla f(x_i, \theta_t)\|_2^2 \cdot \mathbb{E}(|S_b| \,|\, x_i \in S_b))}{p_i} < \sum_{i=1}^{N} \|\nabla f(x_i, \theta_t)\|_2^2 \tag{6}$$

*Proof.* The trace of covariance of regular SGD is

$$Tr(\Sigma(Est')) = \frac{1}{N} \sum_{i}^{N} \|\nabla f(x_i, \theta_t)\|_2^2 - \frac{1}{N^2} (\sum_{i=1}^{N} \|\nabla f(x_i, \theta_t)\|_2)^2. \tag{7}$$

((a)) YearPredictionMSD      ((b)) Slice      ((c)) UJIIndoorLoc

((d)) YearPredictionMSD      ((e)) Slice      ((f)) UJIIndoorLoc

Figure 7: In subplots (a)(b)(c), the comparisons of Wall clock testing loss convergence are made between LGD+adaGrad (red lines) and SGD+adaGrad (blue lines) separately in three datasets. We can see the big gap between them representing LGD converge faster than SGD in time wise. Subplots (d)(e)(f) show the results for same comparisons but in epoch wise.

By 5 and 7, one can easily derive that $Tr(\Sigma(Est)) < Tr(\Sigma(Est'))$ when 6 satisfies.      □

### 3.0.1 LGD for Logistic Regression

Similarly we can derive LGD for logistic regression. The loss function of logistic regression can be written as,

$$L(\theta) = \frac{1}{m} \sum_{i=1}^{m} ln(1 + e^{-y_i \theta x_i}) \tag{8}$$

and the gradient is,

$$\nabla L(\theta)_i = -\frac{x_i y_i}{e^{y_i \theta x_i} + 1}$$

$$\|\nabla L(\theta)_i\| = \frac{\|x_i\|}{e^{y_i \theta x_i} + 1} = \frac{1}{e^{y_i \theta x_i} + 1}$$

Therefore,

$$\operatorname*{argmax}_{i} \frac{1}{e^{y_i \theta^T x} + 1} = \operatorname*{argmax}_{i} -y_i \theta x_i \tag{9}$$

We can save $y_i x_i$ in the hash tables and use $-\theta_i$ as query.

## 4 More Details for Experiment Section

Linear regression experiments used the same three large regression dataset, in the area of musical chronometry, clinical computed tomography, and WiFi-signal localization, respectively. For deep learning experiments, two NLP benchmarks were used. The dataset descriptions and our experiment

results are as follows:

**YearPredictionMSD:** (Lichman, 2013) The dataset contains 515,345 instances subset of the Million Song Dataset with dimension 90. We respect the original train/test split, first 463,715 examples for training and the remaining 51,630 examples for testing, to avoid the 'producer effect' by making sure no song from a given artist ends up in both the train and test set.

**Slice:** (Lichman, 2013) The data was retrieved from a set of 53,500 CT images from 74 different patients. It contains 385 features. We use 42,800 instances as training set and the rest 10,700 instances as the testing set.

**UJIIndoorLoc:** (Torres-Sospedra et al., 2014) The database covers three buildings of Universitat Jaume I with 4 or more floors and almost 110,000 $m^2$. It is a collection of 21,048 indoor location information with 529 attributes containing the WiFi fingerprint, the coordinates where it was taken, and other useful information. We equally split the total instances for training and testing.

**MRPC:** (Dolan & Brockett, 2005) . The dataset is a corpus of sentence pairs automatically extracted from online news sources. It has 4,078 instances with $90\%$ training and $10\%$ validation data.

**RTE:** (Wang et al., 2018) . Each example in these datasets consists of a premise sentence and a hypothesis sentence, gathered from various online news sources. The task is to predict if the premise entails the hypothesis. It has 2,769 instances with $90\%$ training and $10\%$ validation data.

## 5  Implementation details for the experiments on BERT

BERT: Bidirectional Encoder Representations from Transformers uses unidirectional language models for pretraining. For sequence level classification task, BERT can output the final hidden state for the first token in the input as the fixed-dimensional pooled representation of the input sequence. Then for the fine-tuning task, we only need to add an extra classification layer to the BERT model and retrain the overall model jointly. To adapt LGD in fine-tuning tasks in BERT, the fixed-dimensional pooled representation output by BERT pre-trained model are pre-processed in LSH hash tables. During the fine-tuning process, the representations do not change drastically in every iteration so we can periodically update them in hash tables. Furthermore, similar to LGD in linear regression tasks, the parameters in the classification layer are used as queries for sampling the next batch samples.