[Reviews · NeurIPS 2019]

Reviewer 1



Summary: This paper develops a new method for adaptively sampling training examples during stochastic optimization. It is known that the optimal distribution that minimizes the nuclear norm of the covariance of the gradient estimate is one where the the probability of sampling an example is proportional to the magnitude of the gradient of the loss on that example. Sampling according to this distribution is of course impractical, because computing this distribution is as expensive as computing the full gradient and requires O(N) time per iteration. To get around this, prior work either maintains a fixed distribution across all iterations or makes strong assumptions on the distribution of gradients of different training examples (e.g.: the gradients of training examples of the same class are similar). This paper proposes a method that can adaptively sample from different distributions every iteration and requires little assumptions on the distribution of gradients, and yet requires the same per-iteration cost as SGD. This is an important contribution, since conventional wisdom suggests that it is not possible to maintain a distribution over individual training examples from which to sample with a per-iteration cost of less than O(N). (This is referred to as the chicken-and-egg loop in the paper.) This paper shows this is in fact possible for some objective functions and therefore introduces a new class of approaches that can adaptively reduce the variance of gradient estimates without requiring an increase in the per-iteration computational cost. My main suggestion is to more precisely characterize conditions under which the variance of the proposed gradient estimate is smaller than the variance of gradient estimate under uniform sampling. Specifically, p_i should should be upper or lower bounded with the p_1 (true positive probability) and p_2 (false positive probability) associated with the family of hash functions. Then, in the expression for the nuclear norm of the covariance of the gradient estimate, each term should fall into one of three cases: when the gradient magnitude >= t, when it is <= ct, where 0 < c < 1, or when it is between ct and t. Applying the definition of the hash family should give an upper bound on the nuclear norm of the covariance. This should reveal the dependence on p_1 and p_2; one can further optimize over the hyperparameters K and L, which would make dependence of per-iteration cost on p_1, p_2 and the distribution of gradient magnitudes explicit. This should also give sufficient conditions on the distribution of gradient magnitudes (in terms of the proportion of training examples whose gradient magnitudes fall in [0, ct], (ct, t) and [t, inf)) such that the proposed method achieves a lower variance than the uniform sampling scheme. If we then consider the sufficient conditions to keep the nuclear norm of the covariance upper bounded by a constant, this might reveal a dependence on N, and it would be interesting to see what the dependence is. In Sect. 2.3, a few quantities in the expressions for the gradient and the nuclear norm of the covariance are random variables, which makes the expressions random. For example, l is random, because it depends on the sampling over hash tables. Similarly, |S| is random, because it depends on which hash table with a non-empty bucket containing the current value of the parameters is the first to be selected. The randomness for these quantities should be marginalized out to arrive at deterministic expressions, or if this is analytically difficult, high-probability bounds should be derived: to eliminate the randomness in l, you can derive a result of the form "with probability of 1 - \delta, the nuclear norm is upper bounded by some expression, where the probability is over the randomness of sampling of hash tables", and to eliminate the randomness in |S|, it might suffice to consider the worst case, i.e. the largest possible bucket across all parameter values/queries and across all hash tables. Also, the definition of S doesn't seem right, because x doesn't appear anywhere in the expressions for the gradient estimate or the nuclear norm of the covariance. There are more recent nearest neighbor search algorithms that outperform LSH, e.g. DCI [a]. I suspect a sampling version could also be derived, and because MIPS can be reduced nearest neighbor search, such an approach could also be used in the setting considered in this paper. In fact, I believe it could be especially useful here, because DCI does not perform discretization/space partitioning, and so there wouldn’t be the issue of empty buckets. It would be interesting to see how the proposed approach compares to such a baseline. [a] Li & Malik, "Fast k-nearest neighbour search via prioritized DCI", ICML 2017 L232: "by setting small values of K and l" - one does not have the freedom to set l to a particular value, because it is a random variable. What you mean is that by setting K small, l is small with high probability. L233: "if several terms in the summation satisfy |S| / (p_i * N) <= 1" - should this inequality be strict? Also, the proof for this statement should be shown. Minor issues: Eqn. 4 and L123: notation is confusing - angle brackets are typically used to denote inner products rather than concatenation. I would suggest changing them to round brackets. L135: "due to monotonicity" - monotonicity is not enough; you need f to be strictly increasing. If we consider a strictly decreasing function or a constant function (which is non-decreasing), this obviously would not work. L144: "There are few more technical subtleties" - these subtleties should be explained. Also, it should be "a few". Alg. 1: K does not appear in the pseudocode and its meaning is not explained. I assume it's the K from LSH. Also, the notation for collision probability, cp, is a bit confusing, since at first glance it seems to mean some constant some probability p multiplied by some constant c. I would recommend changing it to a single letter. L146: "T is the corresponding feature expansion transformation" - it should be noted that the number of features after expansion grows quadratically in the original dimensionality of the training examples and parameters. Whether or not this introduces complications should be discussed. L186-187: "we can always choose K small enough so that empty buckets are rare". It should be noted that this comes at the cost of having more false positives, i.e. instances where the gradient is actually small in magnitude but gets sampled fairly frequently, because essentially the partitioning of the space is made coarser when K is small. L201: "It is still sub-linear in N (not constant)" - I am not sure if this is a strong argument because making the sampling time constant rather than sub-linear could conceptually make the variance of the gradient estimate increase as N increases, which would lead to slower convergence/more iterations. If the authors believe otherwise, I would suggest analyzing the the variance of the gradient estimate as N increases and showing that the variance is constant in N, perhaps under certain assumptions on the distribution of gradients (as mentioned above). L245: "linear regression and deep learning models" - should clarify whether the deep learning model in question is for regression or classification.

Reviewer 2



Overall I think the work is interesting but there are problems which make it lower than the NeurIPs standard. - I think the experimental results are not convincing. The major contribution of the paper is s fast sampling scheme, mostly derived from regression problem. So the verify the effectiveness, it should work well on all different types of gradient based training. Comparing only to SGD is not valid enough to claim the scheme works. Comparing on more different types of ML task is also necessary. In particular, BERT is well known for unstable fine-tuning stage so comparing on fine-tuning downstream tasks are not convincing. If the task is to compare different methods on pre-training stage of BERT and have a significant gain, it will be more impressive. - From the point of sampling scheme based on the LSH similarity search, I think it makes sense to compare all different methods in [1] and [2] instead of just LSH. Discussing all sort of methods make the present work more complete and have more research value. [1]https://arxiv.org/abs/1806.04189 [2]https://papers.nips.cc/paper/7129-a-greedy-approach-for-budgeted-maximum-inner-product-search.pdf

Reviewer 3



The idea of incorporating hashing techniques for speeding up adaptive sampling in SGD is interesting. However, if my understanding is correct, the proposed scheme heavily relies on the requirement that the gradient takes the form as a nonlinear transformation of the inner product between the parameter (or augmented by some constant) and a per observation-dependent vector. Therefore, the only example in the paper is linear regression (although the authors mentioned in a short paragraph that the method can also be applied to logistic regression. This raises the concern of how useful this method is for solving general problems. In addition, some arguments in the paper do not seem correct. For example, in line 124, the method assumes each feature vector x_i to be normalized. This does not seem possible since in linear regression, the design can only be normalized per dimension of the feature, not per data point. In addition, the calculation in Theorem 2 just displays the resulting variance, and it is clear when this variance would be smaller than the variance with uniform sampling. I noticed that in the supplement, lemma 1 provides a sufficient condition for so. However, the condition is still very complicated and not easy to verify. Last but not least, I tried to read some proofs in supplementary material, but failed due to bad proofreading. For example, the proof of Lemma 1 refers to equation (16) and (17), but there are only 11 labelled equations in the supplement and 5 in the main paper.

[Author Response · NeurIPS 2019]

**To Reviewer 1 and 3:** We are very very fortunate to receive such detailed suggestions that we can bound the nuclear
form of the variance and analyze the condition under which the variance of LGD would be better than that of SGD
from the reviewer who fully understands our work!! The direction the reviewer pointed to is absolutely the right way
to do the analysis, including bounding random variable |S| which itself in expectation is a summation of all collision
probabilities. The analysis for bounding variance is still little involved and will require tools from [1] (like holders
inequality), which we cited at line 235 in our paper. We will add a discussion in the subsequent version of the paper.

A sneak peek of the summary: Our LGD can be viewed as a Kernel Density Estimation(KDE) defined in Equation (1)
from citation [1]. From Lemma (3) of [1], we can similarly get rid of our |S| term by applying Bayes rule to condition
on the random variable $|S|$. Let $p_1 \geq p_2 \geq ... \geq p_N$,

$$\frac{1}{N^2}(\sum_{i=1}^{N} \frac{\|\nabla f(x_i, \theta_t)\|_2^2 \cdot |S|}{p_i} - \frac{1}{N^2}\|(\sum_{i=1}^{N} \nabla f(x_i, \theta_t))\|_2^2 \leq \frac{1}{N^2}(\sum_{i=1}^{N} \frac{\|\nabla f(x_i, \theta_t)\|_2^2}{p_i}(i + \sum_{j=i+1}^{N} \frac{p_j}{p_i}) - \frac{1}{N^2}\|(\sum_{i=1}^{N} \nabla f(x_i, \theta_t))\|_2^2.$$

We denoted the expectation of SGD or LGD as $\mu$. [1] first showED that the trace of the covariance of SGD is tight up
to constants in the worst case, $\mu^2 * \mu^{-1} - \frac{1}{N^2}\|(\sum_{i=1}^{N} \nabla f(x_i, \theta_t))\|_2^2$. The definition of $(\beta, M)$-scale free estimator is
in Definition(3) and that of $(\tau, \gamma)$-localized query is in Definition(5) of the paper [1]. We will then see that the upper
bound on the variance improves when most of the contribution to $\mu$ comes from relatively large gradient norms, which
gives an intuition about when the variance is better. Let LGD be a $(\beta, M)$-scale free estimator with $\beta \in [1/2, 1]$. For
every $(\tau, \gamma)$-localized query $\theta_t$, the upper bound of the trace of the covariance of LGD estimator would be,

$$Tr(\Sigma(Est)) \leq \mu^2 * M^3\{2\gamma^\beta + \gamma^{2-\beta} + \tau^{2\beta-1}\gamma^\beta\}\mu^{-\beta} - \frac{1}{N^2}\|(\sum_{i=1}^{N} \nabla f(x_i, \theta_t))\|_2^2. \tag{1}$$

It also pointed out if there is no assumption made ($\gamma = \tau = 1$), the bound becomes, $\mu^2 * 4M^3\mu^{-\beta} -$
$\frac{1}{N^2}\|(\sum_{i=1}^{N} \nabla f(x_i, \theta_t))\|_2^2$. where the optimal choice of $\beta$ is $\frac{1}{2}$.

We agree with the reviewer that the sampling version of DCI can also be derived. We do not need to aggregate the
samples from all indices to find the nearest neighbor but may just sample several ones similar to LGD. We can easily
add this as a comparison in our paper if needed. We want to point the reviewer to page 5 of our paper that we did
provide empirical analysis and discussions on the change of per-iteration cost, running time and N, K, L, d. We are
also happy to add more thorough experiments for analyzing the effectiveness of all these parameters in the main paper.
We chose Simhash because cosine similarity is useful for the task of inner product of two vectors but we will add a
discussion on choosing teh hash family in the paper.

**To Reviewer 2:** We thank the reviewer for explicitly raising several concerns. However, they are not related to the
paper. Since we think the reviewer has many misunderstanding of the main argument of the paper, we will explain
it here again, and we hope to help the reviewer clarify our goal. The goal of the paper is to provide a better gradient
estimation procedure rather than improving the training of a certain model. SGD is indeed the right baselines as there
is no faster way, in terms of computational cost, to estimate the gradient than random sampling, as mentioned in the
introduction. We have derived the estimation for linear regression, logistic regression, and even neural networks. The
experiment on Bert is not about improving the training accuracy but to show the effectiveness of our superior and fast
estimation of the gradient for faster convergence. Besides, we carefully read both papers referred to in the review, and
we want to clarify that we are not doing near neighbor search. There is **no similarity search with LSH** in our proposal;
it is sampling and unbiased estimation.

We sincerely hope the restatement will help the reviewers become clear about the challenges, motivation, and the goal
of our paper and potentially change the negative view formed due to misunderstanding.

**To Reviewer 3:** We are delighted to see the reviewer's interest in our idea! We want to do a few clarifications, and we
hope the reviewer can have a better understanding of our algorithm and the contribution of the paper. We propose the
first algorithm that achieves sharper estimates of the gradient in near-constant time using hash tables. As a result, we
speed up any first-order gradient-based algorithm, including adagrad, SGD, Adam, etc. Our algorithm uses LSH to
do adaptive sampling, and it is not only restricted to regression tasks. We agree that only for linear models, we can
show connections with optimal sampling. However, for the sampling to be better than random (current practice), we
only want positive correlation with $L_2$ norm of gradient. We can achieve that even by linearizing any non-linear model.
Thus, even for neural networks, we have an informed sampling (it is not optimal, but still better than random sample
based gradient estimation.). We did show the experiments on NN in Section 3.2.

We thank the reviewer for pointing out the problems in our supplementary and give us a chance for a likely score
increase. We apologize for the mislabelling. It should be (11)-(7), (17)->(9), (16)->(8). We will address the typo and
other problems in the main paper and appendix.

[1] Charikar, Moses and Siminelakis, Paris. *Hashing-based-estimators for kernel density in high dimensions* 2017 IEEE
58th Annual Symposium on Foundations of Computer Science (FOCS), 2017.


[Meta-Review · NeurIPS 2019]

This paper received extensive discussion by the reviewers, the meta-reviewer, the SPC, etc. Here is a meta-review summary. The paper considers the problem of adaptively sampling training examples in stochastic optimization, and it shows that it is possible to do so without a per-iteration cost of O(N). This is of interest by itself, since one typically thinks that such sampling requires maintaining a distribution over training examples, which requires O(N) in every iteration, i.e., which is as expensive as full-batch gradient descent. A second aspect of this paper is that the mechanism by which the authors accomplish this is to use LSH, which is a sketching method usually used for nearest neighbor search. Showing that LSH can be used in this way is also interesting by itself since it opens up the possibility to use this method beyond the usual nearest neighbor techniques. On the whole, the paper could be improved (several reviewer comments address this) and there is certainly obvious follow-up work (several reviewer comments also address this), but the current paper as it stands contains a novel combination ideas and methods will be of interest to the NeurIPS community. In particular, the proposed method is applied to SGD, but the basic idea proposed in the paper is orthogonal to particular optimization algorithms and could be applied to momentum, Adam or any other method. Also, while the approach is only derived for linear regression and logistic regression, as pointed out by the authors, it can be extended to models that can be locally linearized and perhaps other extensions in future work. Illustrating the method even in the particular contexts that were used is valuable for the community. After discussion, it was felt that this is the sort of paper that it not an incremental improvement of a paper from last year, but instead combines ideas from several areas in novel ways. While the ideas may take more time to develop fully, the whole point of publishing is to make sure the broader community can build on the early papers and extend the ideas further, and this effort should not be discouraged. Several of the less positive reviews were evaluated by the meta-reviewer, the SPC, etc. in light of that.